# Minimally Invasive Staging of Early-Stage Epithelial Ovarian Cancer versus Open Surgery in Terms of Feasibility and Safety: A Systematic Review and Meta-Analysis

**DOI:** 10.3390/jcm12113831

**Published:** 2023-06-02

**Authors:** Carlo Ronsini, Francesca Pasanisi, Rossella Molitierno, Irene Iavarone, Maria Giovanna Vastarella, Pasquale De Franciscis, Carmine Conte

**Affiliations:** 1Department of Woman, Child and General and Specialized Surgery, University of Campania “Luigi Vanvitelli”, Largo Madonna delle Grazie n.1, 80122 Naples, Italy; carlo.ronsini90@gmail.com (C.R.); molitiernorossella@gmail.com (R.M.); ireneiavarone2@gmail.com (I.I.); mariagiovannavastearella@hotmail.it (M.G.V.); pasquale.defranciscis@unicampania.it (P.D.F.); 2Dipartimento Salute della Donna e del Bambino, Fondazione Universitaria Policlinico A. Gemelli, Largo Agostino Gemelli, 8, 00168 Rome, Italy; carmine.conte87@gmail.com

**Keywords:** ovarian cancer, staging, minimally invasive, laparoscopy

## Abstract

Epithelial ovarian cancer is women’s fourth most common oncological cause of death. One of the main prognostic factors in ovarian cancer is the tumor stage. For instance, surgical staging of the disease is focal when choosing the best therapeutic option for each case. Although open surgery is the prevalent approach for staging and treating ovarian cancer, the use of minimally invasive surgery (MIS) has found recent application in staging or restaging cases of early disease. Our work compares oncological outcomes after MIS staging for FIGO I epithelial ovarian cancer with the laparotomic approach. Following the Preferred Reporting Items for Systematic Reviews and Meta-Analyses (PRISMA) statement recommendations, we systematically searched the Pub Med and Scopus databases in February 2023. No temporal nor geographical limitation was made. We included the articles containing data about Disease-Free Survival (DFS) and Overall Survival (OS), Recurrence Rates (RR), and Upstaging Rates (UpR). We used comparative studies for the meta-analysis. After the database search and article selection, 19 works matched the inclusion criteria for the systematic review. Eleven of these were comparative studies between MIS and Open Surgical Staging (OSS) approaches for ovarian cancer staging and were included in the meta-analysis. The meta-analysis did not show a statistically significant difference between the MIS and the OSS group concerning DFS, OS, and RR. Only Upstaging Rate ≥ FIGO Stage II was statistically significative higher in the OSS group. Likewise, MIS is confirmed to be an approach with a lower profile of surgical complications. In conclusion, our study did not show one approach to be safer than the other. However, the lack of dedicated studies limits the evidence of our study. For instance, we recommend adequately selecting the specimen, minimizing the risk of spillage, and optimizing surgical staging.

## 1. Introduction

Ovarian cancer is the fourth most common oncological cause of death in women worldwide [1]. The main survival prognostic factor in EOC is tumor stage. For instance, patients affected by FIGO stage I EOC have a 5-year survival rate estimated at 70%, as opposed to a FIGO IV survival rate of only 8% [2]. However, early diagnosis of EOC is challenging. Most early diagnoses are accidental and performed by sonography, computerized tomography (CT scanning), or abdominal surgeries for other reasons. Early-stage epithelial ovarian cancer (eEOC) is defined into stages IA, IB, and IC, depending on whether one (stage IA) or both ovaries (stage IB) are involved or if there is capsular rupture, surface tumor, ascites present that contains malignant cells, or positive peritoneal washings (stage IC) [3].

In any case, staging is surgical, and the extent of surgery should be aimed at analyzing structures that might be involved microscopically. Guidelines for staging surgery in early ovarian carcinoma recommend total hysterectomy, bilateral salpingo-oophorectomy, aspiration of cytological washings or ascites, pelvic/para-aortic lymphadenectomy, omentectomy, and multiple peritoneal biopsies performed by the laparotomic route via a longitudinal midline incision [4]. Surgical staging is a focal point to establish the correct diagnosis, through histopathologic evaluation of specimens, and the extent of the disease. Comprehensive staging surgery provides accurate information on prognosis and helps in choosing the best therapeutic option for each case, including adjuvant therapy [4,5]. Although open surgery is the prevalent approach for staging and treating ovarian cancer, in recent years, minimally invasive surgery is becoming applicable not only for treating benign conditions but has also found recent applications in staging or restaging early ovarian disease [6,7]. Several non-randomized studies (NRSs) in early ovarian cancer have reported that laparoscopic surgical staging is a safe and technically feasible procedure [8,9,10,11,12]. The possible advantages of laparoscopy include smaller incisions, less blood loss, faster recovery, shorter hospital stay, fewer complications, less postoperative infection, and better visualization of the peritoneum [13,14,15]. However, laparoscopy has been associated with a higher rate of intraoperative cyst rupture for apparently benign tumors, which may result in upstaging the unexpected ovarian cancer from stage IA or 1B to IC2 [16]. Our systematic review and meta-analysis focuses on analyzing oncological outcomes after minimally invasive surgery (MIS) for the staging of early epithelial ovarian cancer (eEOC), comparing Disease Free Survival (DFS), Overall Survival (OS), Recurrence Rate (RR), and Site of Recurrence between MIS and open surgery staging (LPT) approaches. Secondarily, it also aims to detect the rate of upstaging ≥II FIGO in the two cohorts of patients.

## 2. Materials and Methods

The methods for this study were specified a priori based on the recommendations in the Preferred Reporting Items for Systematic Reviews and Meta-Analyses (PRISMA) statement [17]. We registered the review to the PROSPERO site for meta-analysis with protocol number ID 412599.

### 2.1. Subsection

We performed a systematic search for articles about the surgical staging of EOC in the Pubmed Database and Scopus Database in February 2023. No limitation of publication year was applied. Additionally, we made no geographical restrictions. We considered only studies published entirely in English. Search inputs were TITLE-ABS-KEY (ovarian AND carcinoma) AND TITLE-ABS-KEY (staging) AND TITLE-ABS-KEY (laparoscop*) OR TITLE-ABS-KEY (robot*) OR TITLE-ABS-KEY (laparot*) OR TITLE-ABS-KEY (open AND surgery) OR TITLE-ABS-KEY (minimally AND invasive AND procedures) OR TITLE-ABS-KEY (minimal*) from each database.

### 2.2. Search Method

Study selection was made independently by F.P. and R.M. In case of discrepancy, C.R. decided on inclusion or exclusion. Inclusion criteria were: (1) single-arm studies that included patients with eEOC undergoing staging and treatment by MIS approach; (2) comparative studies that included patients with eEOC undergoing staging and treatment by MIS compared to OSS approach; (3) studies that reported at least one oncological outcome of interest (Disease-Free Survival, DFS; Overall Survival, OS, Recurrence Rate, RR); (3) peer-reviewed articles published originally.

We excluded non-original studies, preclinical trials, animal trials, abstract-only publications, and articles in languages other than English. If possible, the authors of studies that were only published as congress abstracts were attempted to be contacted via email and asked to provide their data. We mentioned the selected studies and all reasons for exclusion in the Preferred Reporting Items for Systematic Reviews and Meta-Analysis (PRISMA) flowchart (Figure 1). We assessed all included studies regarding potential conflicts of interest.

### 2.3. Data Extraction

FP and RM extracted data from all relevant series concerning tumor characteristics, surgical approach, morbidity, and oncological issues such as Recurrences, Deaths, Recurrence Rate (RR), Disease-Free Survival (DFS), Overall Survival (OS), and Upstaging Rate. Additionally, the two groups extracted and compared data about perioperative complications (graded according to the Clavien–Dindo scale) [18]. Disease-Free Survival was defined as the time elapsed between surgery and recurrence. Overall Survival was considered as the time elapsed between surgery and death for disease or the last follow-up. Cancer Recurrence referred to the detection of disease after treatment and after a period of time when the tumor could not be found. Recurrence Rate referred to the percentage of patients from each study that showed cancer recurrence. Upstaging Rate is the percentage of patients that showed a FIGO stage after surgical staging superior to preoperative staging.

### 2.4. Quality Assessment

The quality of the included studies was assessed using the Newcastle–Ottawa scale (NOS) [19]. This assessment scale uses three broad factors (selection, comparability, and exposure), with the scores ranging from 0 (lowest quality) to 8 (best quality). Two authors (R.M. and F.P.) independently rated the quality of the studies. Any disagreement was resolved by discussion or consultation with C.C. We reported the NOS Scale in Appendix A (Table A1). We used a funnel plot analysis to assess publication bias. We used Egger’s regression test to determine the asymmetry of funnel plots.

### 2.5. Statistical Analysis

Heterogeneity among the studies was tested using the Chi-square and I-square tests [20]. The risk rate (RR) and 95% confidence intervals (CI) were used for dichotomous variables. Fixed-effect models were used to conduct statistical analysis without significant heterogeneity (I^2^ < 50%) and random-effect models were used if I^2^ > 50%. DFS and OS were used as clinical outcomes. In each study, Disease-Free Survival was defined as the time elapsed between surgery and recurrence or the date of the last follow-up. Overall Survival has been defined as the time elapsed between surgery and death from disease or the last follow-up. Chi-square tests were used to compare continuous variables. Subgroup analysis in patients undergoing minimally invasive and open surgical staging was performed. Review Manager version 5.4.1 (REVman 5.4.1) and IBM Statistical Package for Social Science (IBM SPSS vers 25.0) for MAC were used for statistic calculation. For all performed analyses, a *p*-value < 0.05 was considered significant.

## 3. Results

### 3.1. Studies’ Characteristics

After the database search, 3515 articles matched the search criteria. After removing records with no full text, duplicates, and wrong study designs (e.g., reviews, case reports), 27 were suitable for eligibility. Of those, 19 articles matched the inclusion criteria and were included in the systematic review. Eight of them were non-comparative, single-armed studies evaluating the outcomes of interest in patients affected by eEOC undergoing minimally invasive staging and surgery. The other 11 were comparative studies between MIS and OS approaches for ovarian cancer staging and were included in the meta-analysis (Figure 1). Data of interest concerning laparoscopic staging were extracted from the aforementioned studies for this review. Moreover, data from comparative studies were used for the meta-analysis. Table 1 summarizes the studies’ characteristics: geolocation, year range, study design, characteristics of the population and sub-cohorts, mean follow-up (FUP), and the number of participants [8,10,11,21,22,23,24,25,26,27,28,29,30,31,32,33,34,35,36]. The quality of all studies was assessed by the Newcastle–Ottawa Scale (NOS) [19] (Appendix A). Overall, the first publication was from 2003. In total, 4041 patients who underwent minimally invasive surgical staging were enrolled. Follow-up ranged from 17 to 84 months on average. Data are summarized in Table 1.

### 3.2. Outcomes

A total of 4041 patients were included in the review. Any of the selected studies presented at least one oncological outcome of interest: DFS and OS at 3 or 5 years and RR in the MIS group. The rate of upstaging >/= II FIGO cases was also evaluated. Moreover, for the meta-analysis, apart from the aforementioned, the total number of complications (intra- and postoperative) was compared in the two groups.

Table 2 refers to the Oncological Outcome of interest only in patients undergoing minimally invasive staging. Only one article [27] did not report DFS or OS outcomes.

The mean 3Y-DFS is 92.1% and 5Y-DFS is 87.1%. Only Park 2018 [11] showed a 3Y-DFS of 100%. Muzii 2018 [25] and Colomer 2008 [8] reported a 95.6% and 95% 3Y-DFS, respectively. A 3Y-DFS of 91.6% is described in Nezhat 2008 [10] and Tozzi 2003 [26]. In Ghezzi 2011 [23], the 3Y-DFS is 91.2%. In Gallotta 2016 [29], the 3Y-DFS is stated at 89%. No data at 5 years are reported in all the aforementioned studies. Merlier 2020 [34] showed a 3Y-DFS and 5Y-DFS of 93% and 88%, respectively. In Ditto 2016 [28], the 3Y-DFS is 90% and 5Y-DFS lowers to 83%. The lowest 3Y-DFS is seen in Minig 2016 [35], corresponding to 87%, lowering to 84% at 5 years follow-up, and Koo 2014 [30] stated 86.1% for both 3Y- and 5Y-DFS. A 5Y-DFS of 91.3% is reported in Lu 2016 [32] and 83% in Gallotta 2021 [22]. Wu 2009 [36] reported the lowest Y5-DFS, stated at 69.5%, and 5Y-OS, stated at 67.4%.

In total, 3Y-OS is the most frequent oncological outcome described overall. The mean 3Y-OS is 97.7% from 15 articles. However, 5Y-OS is described in nine studies and it is estimated at 91.7%. A 100% OS at 3 years is reported in six studies [8,11,25,26,32]. In four studies, the 3Y-OS and 5Y-OS correspond: in Ditto 2016 [28] and Lee 2017 [24] it is stated at 95% for the two articles; in Minig 2018 [35], the OS is 97.3%; and, lastly, Merlier 2020 [34] reported a 97.3% OS. Liu 2013 [31] and Ghezzi 2011 [23], respectively, show 3Y-OS at 97.1% and 97%. In Melamed 2016 [33], 3Y-OS is 94.1% and it is 92% in Gallotta 2016 [29]. Only 5Y-OS is reported in Facer 2019 [21] and Gallotta 2021 [22] at 86.4% and 93.8%, respectively. In Lu 2016 [32], OS at 5 years lowers to 92.9%.

Recurrence rates are described in Table 3. In seven studies, MIS staging data are subjected to OSS with no statistically significant difference. Overall, RR in the MIS group is stated at 7.5%, with the highest RR reported in Gallotta 2021 [22] and Lu 2016 [32], corresponding at 15.3% and 13%, respectively; 12% RR is shown in Minig 2016 [35]; 8.3% RR is stated in Gallotta 2016 [29], Koo 2014 [30], Lee 2017 [24], and Nezhat 2008 [10]. The lowest RRs are reported in Ghezzi 2011 [23] (7.3%), Liu 2013 [31] (5.7%), Merlier 2020 [34] (5.4%), Colomer 2008 [8] (5%), and Muzii 2008 [25] (4.3%). Only in Park 2018 [11], were no recurrences reported both in the MIS and OSS groups.

Table 4 describes the Upstaging Rates in the MIS group and compares them with the OSS group if reported. Six comparative studies [11,28,31,32,33,35] confronted this outcome in the two groups and Melamed 2016 [33] is the only study in which a statistically significant difference is described (*p* < 0.001) with a higher rate of upstaged patients in the OSS cohort (19.2% vs. 12.2%). The highest UpR in the MIS group is 26% in Muzii 2008 [25]. In descending order it was: 25.6% in Ghezzi 2011 [23], 24% in Minig 2016 [35], 21.4% in Lu 2016 [32], 21.1% in Park 2018 [11], and 20% in Colomer 2008 [8] and Ditto 2016 [28]. The lowest UpRs are reported in Nezhat 2008 [10] (19.4%), Gallotta 2021 [22] (18.1%), and Liu 2013 [31] (17.1%), up to 12.2% and 11.3% in Melamed 2016 [33] and Facer 2019 [21], respectively.

**Table 2 jcm-12-03831-t002:** Oncological Outcomes of patients undergoing laparoscopy.

Name	3Y DFS * (%)	3Y OS ° (%)	5Y DFS * (%)	5Y OS ° (%)
Colomer 2008 [8]	95	100	/	/
Ditto 2016 [28]	90	95	83	95
Facer 2019 [21]	/	/	/	86.4
Gallotta 2016 [29]	89	92	/	/
Gallotta 2021 [22]	/	/	83	93.8
Ghezzi 2011 [23]	91.2	97	/	/
Koo 2014 [30]	86.1	/	86.1	/
Lee 2017 [24]	/	95	/	95
Liu 2013 [31]	/	97.1	/	/
Lu 2016 [32]	/	100	91.3	92.9
Melamed 2016 [33]	/	94.1	/	/
Merlier 2020 [34]	93	97.3	88	97.3
Minig 2016 [35]	87	98	84	98
Muzii 2008 [25]	95.6	100	/	/
Nezhat 2008 [10]	91.6	100	/	/
Park 2018 [11]	100	100	/	/
Tozzi 2003 [26]	91.6	100	/	/
Wu 2009 [36]	/	/	69.5	67.4

* Disease Free Survival. ° Overall Survival.

**Table 3 jcm-12-03831-t003:** Recurrence Rate.

Name	Group MIS Recurrence Rate (%)	Group OSS Recurrence Rate (%)	*p*
Colomer 2018 [8]	5	NA	
Gallotta 2016 [29]	8.3	16.3	0.651
Gallotta 2021 [22]	15.3	NA	
Ghezzi 2011 [23]	7.3	NA	
Koo 2014 [30]	8.3	3.8	0.585
Lee 2017 [24]	8.3	NA	
Liu 2013 [31]	5.7	5.0	>0.05
Lu 2016 [32]	13	13	
Merlier 2020 [34]	5.4	29	0.08
Minig 2016 [35]	12	12	0.785
Muzii 2008 [25]	4.3	NA	
Nezhat 2008 [10]	8.3	NA	
Park 2018 [11]	0	0	
Tozzi 2003 [26]	8.3	NA	

NA: Not Available.

**Table 4 jcm-12-03831-t004:** Upstaging Rate ≥II Figo.

Name	Group MIS%	Group OSS%	*p*
Colomer 2018 [8]	20	/	/
Ditto 2016 [28]	20	26	0.63
Facer 2019 [21]	11.3	/	/
Gallotta 2021 [22]	18.1	/	/
Ghezzi 2011 [23]	25.6	/	/
Liu 2013 [31]	17.1	22.5	NA
Lu 2016 [32]	21.4	20.0	0.86
Melamed 2016 [33]	12.2	19.2	<0.001
Minig 2016 [35]	24	14	0.173
Muzii 2008 [25]	26	/	/
Nezhat 2008 [10]	19.4	/	/
Park 2018 [11]	21.1	21.2	0.989

NA: Not Available.

### 3.3. Meta-Analysis

Ten studies comparing outcomes of interest in patients undergoing minimally invasive surgical staging and open surgical staging were enrolled in the meta-analysis. A total of 3278 patients were analyzed. We compared 1467 patients in the MIS arm with 1811 patients who underwent open surgical staging, exploring the risk of recurrences, deaths, complications, and upstaging. Because of low heterogeneity, in all the analyses, a fixed-effects model was applied.

The MIS group was found to be equivalent to the OSS group in terms of risk of recurrence (RR 0.93 [95% CI 0.80–1.09]) (I^2^ = 26%; *p* = 0.21). Those results were not statistically significant (*p* = 0.37) (Figure 2).

Additionally, the two groups were also comparable in death risk (MIS RR 0.85 [95% CI 0.63–1.14]) (I^2^ = 38%; *p* = 0.10). Those results were not statistically significant (*p* = 0.28) (Figure 3).

The complication risk was statistically significantly lower in the MIS group compared to open surgery (MIS RR 0.67 [95% CI 0.50–0.91]; *p* = 0.009) (I^2^ = 28%; *p* = 0.20) (Figure 4).

Finally, the OSS group showed a statistically significantly higher probability to receive an upstaging ≥ FIGO Stage II (MIS RR 0.70 [95% CI 0.57–0.87]; *p* = 0.009) (I^2^ = 37%; *p* = 0.16) (Figure 5).

**Figure 2 jcm-12-03831-f002:**
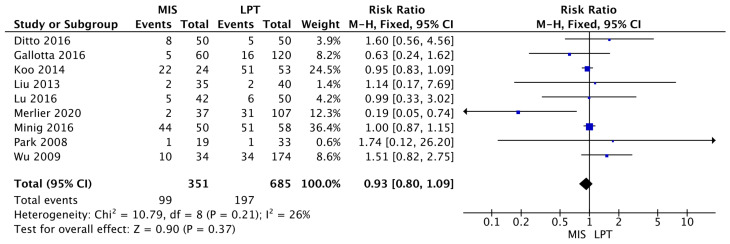
Recurrence Risk [11,28,29,30,31,32,34,35,36].

**Figure 3 jcm-12-03831-f003:**
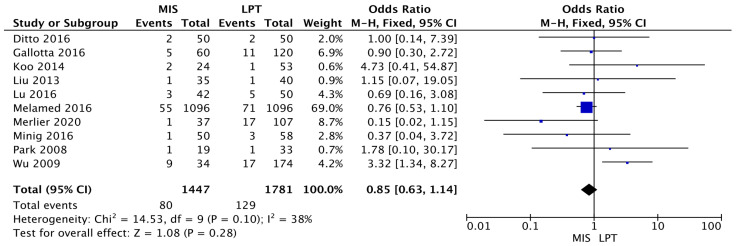
Death Risk [11,28,29,30,31,32,33,34,35,36].

**Figure 4 jcm-12-03831-f004:**
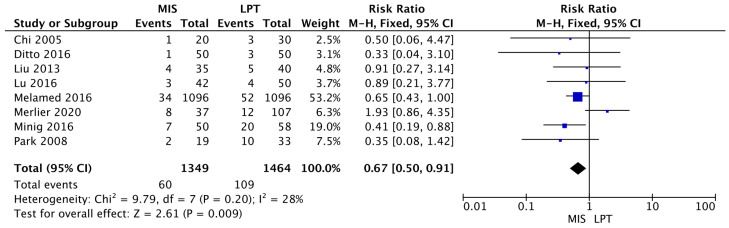
Complication Risk [11,27,28,31,32,33,34,35].

**Figure 5 jcm-12-03831-f005:**
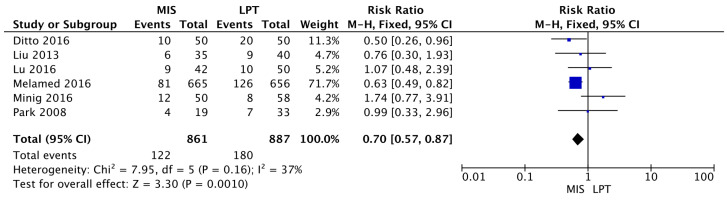
Upstaging [11,28,31,32,33,35].

## 4. Discussion

The management of eEOC has several critical issues. First, the identification of an early stage. To date, no imaging or laboratory test can diagnose the stage of the disease with absolute accuracy. Various hypotheses have followed over the years. The most popular methods are ultrasound, which does not exceed 80% sensitivity [37]. Moreover, this value is drastically reduced when the diagnostic effort focuses on diagnosing the histotype of the lesion as well. Such information would also condition surgical extension, minimizing morbidity; however, to date, this possibility is offered only by the Frozen Section. Good future perspectives come from deepening the knowledge of the tumor microenvironment by performing liquid biopsy [38]. These results appear promising but are still not reproducible in clinical practice. Therefore, the gold standard of staging remains surgery. In light of this, our meta-analysis investigated what the impact of different approaches on safety and oncologic outcomes may be. In the history of gynecologic oncology, we have learned how the same procedure can have different outcomes by laparotomy or minimally invasive approaches. A recent example is what happened with cervical cancer following the LACC Trial [39,40]. This study presented us with the dangers associated with pneumoperitoneum and the potential spread of tumor cells related to circulating CO2 [41]. In parallel, choosing an approach that allows safe manipulation of tumor masses is critical, avoiding intraoperative spillage of tumor material and thus causing iatrogenic upstaging [42]. On the other hand, surgery must ensure proper peritoneal and lymph node staging of the pathology. In the apparent early stage, ovarian carcinoma is associated with occult metastases in other districts in 20–30% [43]. Upstaging of the disease involves not only prognostic impact but also the possibility of the use of other adjuvant and maintenance therapies, such as PARP inhibitors and bevacizumab, depending on mutational status [44,45]. Finally, microscopic lymph node positivity stands at 14% in the literature, and completeness of surgical staging independently impacts patient prognosis [46]. Furthermore, this study demonstrated adjuvant therapy’s greatest prognostic impact in inadequately staged patients. The surgical complexity of this major surgery is hypothesized to be greater in minimally invasive approaches. Therefore, we investigated how MIS and open approaches may impact prognosis. There are critical issues that need to be considered. Notoriously, the MIS approach has been associated with a higher risk of spillage. In 2020, Matsuo et al. proved an increased risk of tumor capsule rupture with MIS in a large retrospective analysis of 8850 early ovarian cancer patients and the deleterious oncological influence [47]. In line with this, careful sample selection could minimize this risk, as recently demonstrated by Ghirardi et al., who researched preoperative and intraoperative features predictive of spillage [48]. Currently, our group is conducting a prospective study focused on calculating the risk of MIS-related intraoperative spillage based solely on the characteristics of the adnexal neoformation (Ovarian Cyst’s Enucleation Spillage Score OC-ESS NCT05376384). Another risk associated with the MIS approach is the development of port-site metastasis, but this remains a rare occurrence, with an incidence of ≤2% [49,50,51]. Nevertheless, our data did not show a clear advantage of one approach over the other, concerning oncologic outcomes. This is probably because the impact of the surgical approach is minimized by intrinsic features of the tumor, such as high grade and mutational status, wherein the appropriateness of adjuvant therapy affects prognosis. This hypothesis is also confirmed by Bentivegna et al., who demonstrated with a meta-analysis including 1150 patients in 32 different papers how the greatest prognostic impact is related to a high grade [52]. Another parameter to consider is the type of recurrence related to the approach. Hypothetically, circulating CO2 during minimally invasive approaches could promote the extra-pelvic spread of pathology and result in distant metastasis. Our work showed no differences in terms of recurrence rate and type of recurrence. This is probably related to the pathology’s nature and propensity to impart peritoneal carcinosis. A concordant finding was demonstrated by Bentivegna et al. in 2016. In this extensive review on eEOC fertility-sparing treatments, they reported how even in the presence of the ovary site, 90% of recurrences are extra-pelvic in high-grade cases [52]. Therefore, with equal prognostic impact, it is appropriate to consider the safety profile of the surgical approach. Our meta-analysis confirmed that MIS approaches have a lower risk of severe complications than LPT. This evidence was already confirmed in the meta-analysis of Bogani et al. [53]. Minimally invasive surgical staging in early stage ovarian carcinoma: a systematic review and meta-analysis, showing that the rate of postoperative complications and risk of transfusions was higher in patients undergoing open surgery compared to MIS. Another interesting point is that our manuscript also demonstrated superior power on the part of the open approach to staging patients (MIS RR 0.70 [95% CI 0.57–0.87]; *p* = 0.009) (I^2^ = 37%; *p* = 0.16). Likely, this can be attributed to the greater ease of performing complicated surgeries such as lumbo-aortic lymphadenectomy and better management of the upper abdomen, with a likely greater extent of omentectomy, compared to pelvic surgery which is also easily performed with the MIS approach [54]. Moreover, the risk of tumor spillage via MIS, one of the greatest causes of upstaging in ovarian cancer I FIGO stage, can be controlled if we consider that these surgeries are performed in highly specialized centers by surgeons with extensive experience. Then, MIS staging may be performed in patients with specific tumor features; for instance, as already mentioned above, MIS staging must be performed on a selected sample of patients with specific features, minimizing the risk of spillage [48]. Introducing the sentinel lymph node in managing ovarian cancer could partially overcome these limitations. To date, this practice remains exclusively experimental [55,56]. Undoubtedly, our study is limited by the paucity of randomized clinical trials on the subject, although it enjoys a solid sample size given the numerous retrospective studies. Another limitation is the inability to minimize the variety of results related to individual surgeons, different experience levels, and skills. Several systematic reviews and meta-analyses have been carried out on the topic over the years [53,57]. Bogani et al. [53] completed a comprehensive review in 2017; its findings regarding the oncological outcomes are confirmed in our work, except for the Upstaging Rate, which is shown to be equivalent in the two groups. This review points out an important issue, which is that laparoscopic staging is associated with a shorter time to chemotherapy than laparotomic procedures, which surely impacts prognosis. Kong Q et al. 2020 [57] is the most recent review about ovarian cancer OSS vs. MIS staging. It shows no statistical differences in terms of OS, PFS, and UpR; only RR is shown to be lower in the MIS approach. However, this work uses only six studies that were overall eligible for the meta-analysis. In our work we made an extremely accurate selection of the articles suitable for inclusion considering the outcomes we wanted to investigate, the quality of the works, and studies being performed in high-volume centers, managing ovarian pathology as the main disease. Using data taken from the selected studies, we performed a new statistical analysis. We believe MIS requires more expertise than laparotomy, and real evidence will only come from randomized trials conducted in centers of excellence. Likewise, however, the breadth of research allows us to emphasize that a real increased risk profile in the case of MIS does not exist. Thus, it should be up to the individual, given equal surgical performance, to opt for the approach with less morbidity.

## 5. Conclusions

Our meta-analysis did not show that MIS is safer than open techniques or vice versa. The lack of dedicated studies, however, limits the evidence of our study, as we needed a clearer picture of the risks related to the surgeons’ experience. Likewise, MIS is confirmed to be an approach with a lower profile of surgical complications. We believe that, in experienced hands, MIS can be a viable alternative in staging eEOC. However, we recommend adequately selecting the specimen, minimizing the risk of spillage, and optimizing surgical staging.

## Figures and Tables

**Figure 1 jcm-12-03831-f001:**
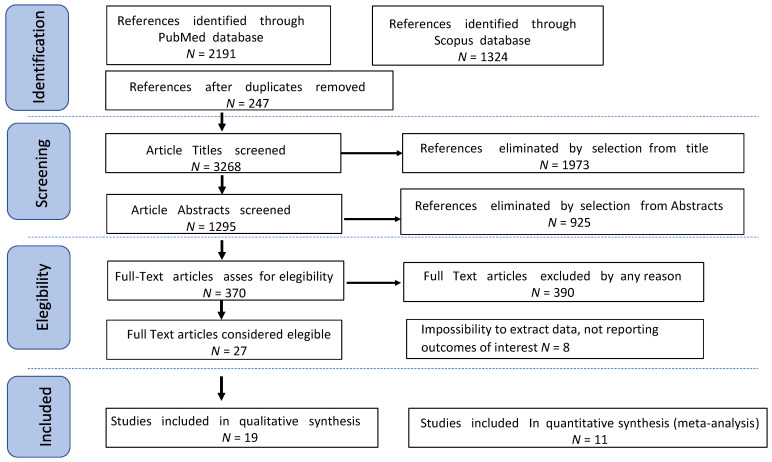
Preferred Reporting Items for Systematic Reviews and Meta-Analysis (PRISMA) flowchart.

**Table 1 jcm-12-03831-t001:** Studies’ characteristics.

Single Arm Studies						
Name	Country	Study Design	Study Year	Population	N of Participant, Total (MIS/OSS) ^	Mean FUP Months
Colomer 2008 [8]	Spain	Prospective Monocentric study	2003–2007	Apparent early-stage epithelial ovarian cancer undergoing MIS staging	19	24.7
Facer 2019 [21]	USA	Retrospective Cohort Multicentric Study	2010–2014	Apparent early-stage epithelial ovarian cancer undergoing MIS staging	1901	37.6
Gallotta 2021 [22]	Italy	Retrospective Observational Multicentric Study	2008–2016	Apparent early-stage epithelial ovarian cancer undergoing MIS staging	254	61
Ghezzi 2011 [23]	Italy	Retrospective Observational Multicentric study		Apparent early-stage epithelial ovarian cancer undergoing MIS staging	82	28.5
Lee 2017 [24]	Taiwan	Retrospective Observational Monocentric study	2002–2014	Apparent early-stage epithelial ovarian cancer undergoing MIS staging	24	31.5
Muzii 2008 [25]	Italy	Prospective Observational Monocentric study	2003–2013	Apparent early-stage epithelial ovarian cancer undergoing MIS staging	27	20
Nezhat 2008 [10]	USA	Retrospective Observational Monocentric study	1995–2007	Apparent early-stage epithelial ovarian cancer undergoing MIS staging	36	55.9
Tozzi 2003 [26]	Germany	Prospective Observational Monocentric study	1996–2003	Apparent early-stage epithelial ovarian cancer undergoing MIS staging	24	46.4
**Comparative Studies**						
Chi 2005 [27]	USA	Retrospective Case-Control Monocentric Study	2000–2003	Apparent early-stage epithelial ovarian cancer undergoing MIS vs. OSS staging	50 (20/30)	46
Ditto 2016 [28]	Italy	Retrospective Case-Control Multicentric study	2005–2015	Apparent early-stage epithelial ovarian cancer undergoing MIS vs. OSS staging	100 (50/50)	51.1
Gallotta 2016 [29]	Italy	Retrospective Case-Control Multicentric study	2000–2013	Apparent early-stage epithelial ovarian cancer undergoing MIS vs. OSS staging	180 (60/120)	38
Koo 2014 [30]	Korea	Retrospective Case-Control Monocentric study	2006–2012	Apparent early stage epithelial ovarian cancer undergoing MIS vs. OSS staging	77 (24/53)	31
Liu 2013 [31]	China	Retrospective Case-Control Monocentric study	2002–2010	Apparent early-stage epithelial ovarian cancer undergoing MIS vs. OSS staging	75 (35/40)	84
Lu 2016 [32]	China	Retrospective Case-Control Monocentric study	2002–2014	Apparent early-stage epithelial ovarian cancer undergoing MIS vs. OSS staging	92 (42/50)	82
Melamed 2016 [33]	USA	Retrospective Observational Multicentric study	2010–2012	Apparent early-stage epithelial ovarian cancer undergoing MIS vs. OSS staging	4798 (1112/3686)	29.9
Merlier 2020 [34]	France	Retrospective Case-Control Multicentric study	2000–2018	Apparent early-stage epithelial ovarian cancer undergoing MIS vs. OSS staging	144 (37/107)	36
Minig 2016 [35]	Spain	retrospective comparative observational study	2006–2014	Apparent early-stage epithelial ovarian cancer undergoing MIS vs. OSS staging	108 (50/58)	30.4
Park 2008 [11]	USA	Retrospective Case-Control Monocentric study	2004–2007	Apparent early-stage epithelial ovarian cancer undergoing MIS vs. OSS staging	52 (19/33)	17
Wu 2009 [36]	Taiwan	Retrospective Case-Control Monocentric study	1984–2006	Apparent early-stage epithelial ovarian cancer undergoing MIS vs. OSS staging	208 (34/174)	65

MIS: Minimally Invasive Surgery; OSS: Open Surgical Staging; FUP: Follow-Up. ^ Sub-analysis of the entire cohort.

## Data Availability

Not applicable.

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
