# Peer review of "Minimally Invasive Staging of Early-Stage Epithelial Ovarian Cancer versus Open Surgery in Terms of Feasibility and Safety: A Systematic Review and Meta-Analysis"

_jcm, 2023, doi:10.3390/jcm12113831_

Round 1

Reviewer 1 Report

Comments to authors:

Surgical staging is one of the most important issue for the treatment and prognosis of ovarian cancer. In this manuscript, Ronsini et al focused on different outcomes by laparotomy or minimally invasive surgery, and compared the oncological outcomes after these different approaches for FIGO I epithelial ovarian cancer by meta-analysis. They found that it did not show a significant difference between the MIS and the open surgery staging concerning DFS, OS, and RR. MIS showed a lower profile of surgical complications and upstaging. The analysis is properly conducted. The findings showed certain novelty and significance, which are interested by the physicians and offered meaningful reference for clinical decision-making. There are some flaws and issues that need further clarification.

1.     The result of the comparison of upstaging was not mentioned in the abstract or the section of discussion. Please include it properly in the manuscript.

2.     It showed in the section of result that “The OSS group showed a statistically significant higher probability to receive an upstaging ≥ FIGO Stage II (MIS RR 0.70 [95% CI 0.57-0.87]; p=0.009) (I2=37%; p = 0.16)”. Tumor spillage is one of the major concern about MIS. However, this study showed opposite result. This is really something that need further discussion. Please include it in the section of discussion.

3.     The section of discussion need improvement by making sufficient discussion on each findings of the study.

Author Response

Dear Reviewer,

Thank You for taking the time to review our manuscript and for your comments. They are crucial and valuable to us in raising the quality standard of our work.

You can read below the reply to each point:

  1. The result of the comparison of upstaging was not mentioned in the abstract or the section of discussion. Please include it properly in the manuscript.

Results on the comparison of upstaging in the two groups has been added to the abstract as suggested (lines 27-28). Discussion has been improved as suggested as well. (see point 3)

  1. It showed in the section of result that “The OSS group showed a statistically significant higher probability to receive an upstaging ≥ FIGO Stage II (MIS RR 0.70 [95% CI 0.57-0.87]; p=0.009) (I2=37%; p = 0.16)”. Tumor spillage is one of the major concern about MIS. However, this study showed opposite result. This is really something that need further discussion. Please include it in the section of discussion

Our study shows that the Upstaging rate is higher in the OSS group. As explained in the discussion, “this can be attributed to the greater ease of performing complicated surgeries such as lumbo-aortic lymphadenectomy and better management of the upper abdomen, with a likely greater extent of omentectomy, compared to pelvic surgery which is easily performed also with MIS approach” (lines 308-311). Moreover, the risk of spillage via MIS can be controlled if we consider that these surgeries are performed in highly specialized centers by surgeons with extensive experience and that MIS staging may be held in patients with specific features (lines 311-317), for instance, as we indicated in the discussion, “careful sample selection could minimize this risk, as recently demonstrated by Ghirardi et al., who researched preoperative and intraoperative features predictive of spillage” (277-280).

  1. The section of discussion need improvement by making sufficient discussion on each findings of the study.

Thank you for your suggestion. We added some more considerations on each findings of the study. The first point concerns oncologic outcomes, and how the prognosis is shown not to be influenced by the type of approach. MIS, apparently,  seems to be associate to a higher risk of tumor spillage then OSS (as demonstrated by some works), leading to an increased risk of recurrences. Nevertheless, our data did not show a clear advantage of one over the other tecnique concerning oncological survival outcomes and recurrence rate. (lines 277- 297). One more focus is on intra and post operative complications, resulting mainly associated to OSS staging then MIS staging. (lines 299-305). Lastly, we report a focus on upstaging rate that our work showed to be higher in OSS approach (308-317).

You can find the rewritten and corrected manuscript version in the attached file. We highlighted any changes made. Moreover,  we have made a general revision of the English vocabulary and grammar thanks to two further authors, I. Iavarone and M.G. Vasterella, that have been added to the list of authors.

Thank you very much for your advice and comments. We hope we have complied with your requests.

Reviewer 2 Report

MIS versus open laparotomy for early stage ovarian cancer is a topic of great interest to gynecologic oncologists, and I appreciate the nice summary of the papers that have been published [1, 2]. The content is very reasonable and should be of sufficient interest to the readers. However, one question I would like to ask is that there are other meta-analysis papers and they do not seem to be very different, especially compared to the one published in Italy in 2017 [1].

I'm asking for further clarification on this. For example, what is different and are there any new conclusions?

[1] Bogani G, Borghi C, Leone Roberti Maggiore U, Ditto A, Signorelli M, Martinelli F, et al. Minimally Invasive Surgical Staging in Early-stage Ovarian Carcinoma: A Systematic Review and Meta-analysis. J Minim Invasive Gynecol. 2017; 24: 552-562.

[2] Kong Q, Wei H, Zhang J, Li Y, Wang Y. Comparison of the survival outcomes of laparoscopy versus laparotomy in treatment of early-stage ovarian cancer: a systematic review and meta-analysis. J Ovarian Res. 2021; 14: 45.

Average to high

Author Response

Dear Reviewer,

Thank You for taking the time to review our manuscript and for your comments. They are crucial and valuable to us in raising the quality standard of our work.

Several systematic reviews and meta-analysis have been conducted on the topic over the years. In our work, we made a highly accurate selection of the articles suitable of inclusion, considering the outcomes we wanted to investigate and the quality of the works, excluding studies by the same authors, realized in different years and using the same samples of patients and including only the most recent and updated ones. We performed a new statistical analysis using data taken from the selected studies. In the discussion, however, we have added to the discussion some more considerations and the other references and their results. (lines 323-336)

You can find the rewritten and corrected manuscript version in the attached file. We highlighted any changes made. Moreover,  we have made a general revision of the English vocabulary and grammar thanks to two further authors, I. Iavarone and M.G. Vasterella, that have been added to the list of authors.

Thank you very much for your advice and comments. We hope we have complied with your requests.
